# SSMF: Shifting Seasonal Matrix Factorization

**Koki Kawabata** [*]
SANKEN Osaka University
koki@sanken.osaka-u.ac.jp

**Siddharth Bhatia** [*]
National University of Singapore
siddharth@comp.nus.edu.sg

**Rui Liu**
National University of Singapore
xxliuruiabc@gmail.com

**Mohit Wadhwa**
mailmohitwadhwa@gmail.com

**Bryan Hooi**
National University of Singapore
bhooi@comp.nus.edu.sg

## Abstract

Given taxi-ride counts information between departure and destination locations, how can we forecast their future demands? In general, given a data stream of events with seasonal patterns that innovate over time, how can we effectively and efficiently forecast future events? In this paper, we propose **S**hifting **S**easonal **M**atrix **F**actorization approach, namely SSMF, that can adaptively learn multiple seasonal patterns (called regimes), as well as switching between them. Our proposed method has the following properties: (a) it accurately forecasts future events by detecting regime shifts in seasonal patterns as the data stream evolves; (b) it works in an online setting, i.e., processes each observation in constant time and memory; (c) it effectively realizes regime shifts without human intervention by using a lossless data compression scheme. We demonstrate that our algorithm outperforms state-of-the-art baseline methods by accurately forecasting upcoming events on three real-world data streams.

## 1 Introduction

The ubiquity of multi-viewed data has increased the importance of advanced mining algorithms that aim at forecasting future events [Bai et al., 2019, Wu et al., 2019, Deng et al., 2020], such as forecasting taxi-ride counts based on their departure and destination locations, and time information, where the events can be represented as a stream of tuples $(departure, destination, time; count)$. Infection spread rates during epidemics can also be represented in a similar form as $(location, disease, time; count)$, and the task is to forecast the infection count. Given such a large data stream produced by user events, how can we effectively forecast future events? How can we extract meaningful patterns to improve forecasting accuracy? Intuitively, the problem we wish to solve is as follows:

**Problem 1** *Given an event data stream that contains seasonal patterns, forecast the number of future events between each pair of entities in a streaming setting.*

As natural periodicity occurs in several types of data, one of the promising directions to understand the vital dynamics of a data stream is to consider seasonal phenomena. However, high-dimensional data

---

[*]Equal contribution

35th Conference on Neural Information Processing Systems (NeurIPS 2021).

Table 1: Relative Advantages of SSMF

| | SVD/NMF++ | RegimeCast | TRMF | SMF | **SSMF** |
|---|---|---|---|---|---|
| Sparse Data | ✓ | - | ✓ | ✓ | ✓ |
| Regime Shifts | - | ✓ | - | - | ✓ |
| Seasonal Pattern | - | - | ✓ | ✓ | ✓ |
| Drifting components | some | - | - | ✓ | ✓ |

structures in sparse time-series pose many challenging tasks to uncover such valuable seasonal patterns [Chi and Kolda, 2012], while modeling seasonality or periodic patterns has been a well-studied topic in literature [Takahashi et al., 2017]. Moreover, in a streaming setting, data dynamics vary over time, and thus the models learned with historical data can be ineffective as we observe new data. Although online learning schemes, such as [Schaul et al., 2013, Jothimurugesan et al., 2018, Yang et al., 2018, Nimishakavi et al., 2018, Lu et al., 2019], provide solutions to overcome this concern, disregarding all past dynamics fails to capture re-occurring patterns in the subsequent processes. On non-stationary data streams, such approaches no longer converge to the optimal parameters but capture average patterns for several phenomena, which causes less accurate forecasting. Adequate methods should retain multiple patterns where each pattern is updated incrementally for prevailing tendencies and leveraged for forecasting.

In this paper, we focus on online matrix factorization for real-time forecasting with seasonality. Specifically, we propose a streaming method, Shifting Seasonal Matrix Factorization (SSMF), which allows the factorization to be aware of multiple smooth time-evolving patterns as well as abrupt innovations in seasonal patterns, which are retained as *regimes*. In summary, our contributions are:

1. **Accuracy:** Outperforms state-of-the-art baseline methods by accurately forecasting 50-steps ahead of future events.

2. **Scalability:** Scales linearly with the number of regimes, and uses constant time and memory for adaptively factorizing data streams.

3. **Effectiveness:** Finds meaningful seasonal patterns (regimes) throughout the stream and divides the stream into segments based on incrementally selected patterns. SSMF uses a lossless data compression scheme to determine the number of regimes.

**Reproducibility:** Our datasets and source code are publicly available at: `https://www.github.com/kokikwbt/ssmf`

## 2 Related Work

Table 1 summarizes the relative advantages of SSMF, compared with existing methods for matrix factorization and time series forecasting. Our proposed method satisfies all the requirements. The following two subsections provide the details of related work on matrix/tensor factorization and time series modeling.

### 2.1 Matrix Factorization

The major goal of matrix/tensor factorization, well known as SVD, is to obtain lower-dimensional linear representation that summarizes important patterns [Chi and Kolda, 2012], and it has a wide range of applications [Hoang et al., 2017, Yelundur et al., 2019, Tišljarić et al., 2020]. Non-negative constraint provides an ability to handle sparse data and find interpretable factors [Cichocki et al., 2009]. [Takahashi et al., 2017] adapted tensor factorization to extract seasonal patterns in time series as well as the optimal seasonal period in an automatic manner. [Yu et al., 2016] formulated the temporal regularized matrix factorization (TRMF), which is designed for time series prediction. [Song et al., 2017, Najafi et al., 2019] addressed stream tensor completion but didn't extract seasonal patterns. Recently, SMF [Hooi et al., 2019] proposed an online method that incorporates seasonal factors into the non-negative matrix factorization framework, and outperforms CPHW [Dunlavy et al., 2011], which is an offline matrix factorization method. However, SMF misses regime shifts which should be modeled individually.

## 2.2 Time Series Analysis

Statistical models, such as autoregression (AR) and Kalman filter (KF), learn temporal patterns for forecasting, thus a lot of their extensions have been proposed [Cai et al., 2015, Dabrowski et al., 2018]. RegimeCast [Matsubara and Sakurai, 2016] focuses on regime detection to obtain compact factors from data streams effectively but it does not consider updating detected regimes. Also, note that they do not consider mining sparse data in the non-linear dynamical systems, while handling missing values has a significant role in time series analysis [Yi et al., 2016, Abedin et al., 2019]. Hidden Markov models are also powerful tools for change detection in time series [Mongillo and Deneve, 2008, Montanez et al., 2015, Pierson et al., 2018, Kawabata et al., 2019], but these models represent only short temporal dependency and thus are ineffective for forecasting tasks. Detecting regime shifts in non-stationary data can be categorized in Concept Drift, and its theory and recent contributions are well-summarized in [Lu et al., 2019]. In more detail, we consider incremental drift and reoccurring concepts simultaneously. [Wen et al., 2019] tried to decompose seasonality and trends in time series on the sparse regularization framework to handle fluctuation and abrupt changes in seasonal patterns, but this approach does not focus on stream mining [Krempl et al., 2014].

## 3 Proposed Model

In this section, we propose a model for shifting seasonal matrix factorization. A tensor we consider consists of a timestamped series of $(m \times n)$ matrices $\mathbf{X}(1), \ldots, \mathbf{X}(r)$, which can be sparse, until the current time point $r$. We incrementally observe a new matrix $\mathbf{X}(r + 1)$ and $r$ evolves ($r = r + 1$). Our goal is to forecast $\mathbf{X}(t)$ where $r < t$ by uncovering important factors in the flow of data, whose characteristics can change over time. As we discussed the effectiveness of handling seasonal patterns, we incorporate seasonal factors into a switching model that can adaptively recognize recent patterns in forecasting.

### 3.1 Modeling Shifting Seasonal Patterns

Figure 1 shows an overview of our model. In matrix factorization, we assume that each attribute, e.g., a start station where we have a taxi ride, has $k$ latent communities/activities, which is smaller than $m$ and $n$, e.g., the number of stations. Considering such latent seasonal activities, an input data stream is decomposed into the three factors: $\mathbf{U}$ with shape $(m \times k)$ and $\mathbf{V}$ with shape $(n \times k)$ for the row and columns of data $\mathbf{X}(t)$, respectively, and seasonal patterns $\mathbf{W}$ with period $s$, i.e., $(s \times k)$, where $k$ is the number of components to decompose. The $i$-th row vector $\mathbf{w}_i = \{w_1, \ldots, w_k\}$ of $\mathbf{W}$ corresponds to the $i$-th season. Letting $\mathbf{W}_i$ be a $(k \times k)$ diagonal matrix (zeros on non-diagonal entries) with diagonal elements as $\mathbf{w}_i$, $\mathbf{X}(t)$ is approximated as follows.

$$\mathbf{X}(t) \approx \mathbf{U}\mathbf{W}_i\mathbf{V}^T. \tag{1}$$

More importantly, we extend $\mathbf{W}$ by stacking additional $\mathbf{W}$ to capture multiple discrete seasonal patterns as *regimes*, which is shown as the red tensor in Figure 1. Letting $g$ be the number of regimes, our model employs a seasonality tensor $\mathcal{W} = \{\mathbf{W}^{(1)}, \ldots, \mathbf{W}^{(g)}\}$, where each slice $\mathbf{W}^{(i)}$ captures the $i$-th seasonal pattern individually. By allowing all these components to vary over time, $\mathbf{X}(t)$ can be represented as follows.

$$\mathbf{X}(t) \approx \mathbf{U}(t)\mathbf{W}_i^{(z)}(t)\mathbf{V}(t)^T, \tag{2}$$

where $z \in \{1, \ldots, g\}$ shows an optimal regime index to represent $\mathbf{X}(t)$ in the $i$-th season. That is, our approach adaptively selects $\mathbf{w}_i^{(z)}$, as shown in the red-shaded vector in the figure, with regard to the current season and regime. Finally, the full parameter set we want to estimate is as follows.

**Problem 2 (Shifting seasonal matrix factorization: SSMF)** *Given: a tensor stream* $\mathbf{X}(1), \ldots,$ $\mathbf{X}(r)$ *that evolves at each time point,* **Maintain:** *community factors:* $\mathbf{U}(t), \mathbf{V}(t)$, *seasonal factors:* $\mathcal{W}(t)$, *and the number of regime $g$ in a streaming fashion.*

### 3.2 Streaming Lossless Data Compression

In the SSMF model, deciding the number of regimes, $g$, is a crucial problem for accurate modeling because it should change as we observe a new pattern to shift. It is also unreliable to obtain any error

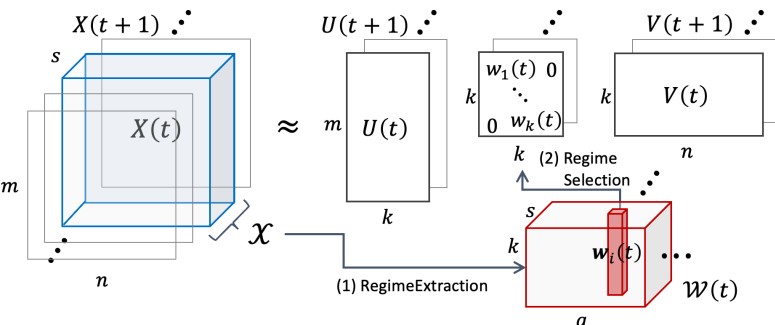

Figure 1: An overview of our model: it smoothly updates all the components, $\mathbf{U}(t), \mathbf{V}(t)$, and $\mathcal{W}(t)$, to continuously describe data stream $\mathbf{X}(t)$. If required, (1) increases the number of regimes, $g$, by extracting a new seasonal pattern in the last season $\mathcal{X} = \{\mathbf{X}(t - s + 1), \ldots, \mathbf{X}(t)\}$. Then, (2) selects the best $\mathbf{w}_i^{(z)}(t)$ in regimes to describe $\mathbf{X}(t)$.

thresholds to decide the time when employing a new regime before online processing. Instead, we want the model to employ a new regime incrementally so that it can keep concise yet effective to summarize patterns. The minimum description length (MDL) principle [Grünwald and Grunwald, 2007] realizes this approach, which is defined as follows.

$$\underbrace{< \mathbf{X}; \mathbf{U}, \mathbf{V}, \mathcal{W} >}_{\text{Total MDL cost}} = \underbrace{< \mathbf{U} > + < \mathbf{V} > + < \mathcal{W} >}_{\text{Model description cost}} + \underbrace{< \mathbf{X}|\mathbf{U}, \mathbf{V}, \mathcal{W} >}_{\text{Data encoding cost}}, \quad (3)$$

where $< \mathbf{X}; \mathbf{U}, \mathbf{V}, \mathcal{W} >$ is the total MDL cost of $\mathbf{X}$ when given a set of factors, $\mathbf{U}, \mathbf{V}$ and $\mathcal{W}$.

Model description cost, $< \cdot >$, measures the complexity of a given model, based on the following equations.

$$< \mathbf{U} > = |\mathbf{U}| \cdot (\log(m) + \log(k) + c_F),$$
$$< \mathbf{V} > = |\mathbf{V}| \cdot (\log(n) + \log(k) + c_F),$$
$$< \mathcal{W} > = |\mathcal{W}| \cdot (\log(g) + \log(s) + \log(k) + c_F),$$

where $|\cdot|$ shows the number of nonzero elements in a given matrix/tensor, $c_F$ is the float point cost[2].

Data encoding cost, $< \mathbf{X}|\cdot >$, measures how well a given model compress original data, for which the Huffman coding scheme [adl, 2001] assigns a number of bits to each element in $\mathbf{X}$. Letting $\hat{x} \in \hat{\mathbf{X}}$ be a reconstructed data, the data encoding cost is represented by the negative log-likelihood under a Gaussian distribution with mean $\mu$ and variance $\sigma^2$ over errors, as follows.

$$< \mathbf{X}|\mathbf{U}, \mathbf{V}, \mathcal{W} > = \sum_{x \in \mathbf{X}} - \log_2 P_{\mu,\sigma}(x - \hat{x}). \quad (4)$$

While we cannot compute the cost with all historical data in a streaming setting, a straightforward approach obtains the best number of regimes so that the total description cost is minimized. To compute the total cost incrementally and individually when provide with new data, we decompose $< \mathcal{W} >$ approximately into:

$$< \mathcal{W} > \approx \sum_{i=1}^{g} < \mathbf{W}^{(i)} > = \sum_{i=1}^{g} |\mathbf{W}^{(i)}| \cdot (\log(s) + \log(k) + c_F). \quad (5)$$

When a new matrix $\mathbf{X}$ arrives, we decide whether the current $\mathcal{W}$ should increase the number $g$. by comparing the total costs for existing regimes with the total costs for estimated regime.

$$\mathbf{W}_{best} = \underset{\mathbf{W}' \in \{\mathbf{W}^{(i)}, \ldots, \mathbf{W}^{(g+1)}\}}{\arg \min} < \mathbf{X}; \mathbf{U}, \mathbf{V}, \mathbf{W}' >, \quad (6)$$
$$= \underset{\mathbf{W}' \in \{\mathbf{W}^{(i)}, \ldots, \mathbf{W}^{(g+1)}\}}{\arg \min} < \mathbf{U} > + < \mathbf{V} > + < \mathbf{W}' > + < \mathbf{X}|\mathbf{U}, \mathbf{V}, \mathbf{W}' >.$$

[2]We used $c_F = 32$ bits.

If $\mathbf{W}^{(g+1)}$ gives the minimum total cost of $\mathbf{X}$, then it belongs to the full $\mathcal{W}$ for subsequent modeling. In other words, the size of $\mathcal{W}$ is encouraged to be small unless we find significant changes in seasonal patterns.

# 4 Proposed Algorithm

We propose an online algorithm for SSMF on large data streams containing seasonal patterns. Algorithm 1 summarizes the overall procedure of our proposed method: it first detects the best regime for the most recent season through a two-step process, namely, RegimeSelection and RegimeExtraction. Then it smoothly updates the entire factors based on the selected regime with non-negative constraints. When deciding the number and assignment of regimes, the algorithm evaluates the cost, Equation (6), for the most recent season, to ensure the smoothness of consecutive regime-switching during a season. Thus, the algorithm keeps a sub-tensor $\mathcal{X}$, which consists of the matrices from time point $t - s + 1$ to $t$, as a queue while receiving a new $\mathbf{X}(t)$. Given a $\mathcal{X}$, the best regime at time point $t$ is obtained as follows.

- RegimeSelection, which aims to choose the best regime $\mathbf{W}_{RS}$ that minimizes Equation (6) in $\{\mathbf{W}^{(1)}, \ldots, \mathbf{W}^{(g)}\}$, using the two previous factors $\mathbf{U}(t-1)$ and $\mathbf{V}(t-1)$. We compute the total cost, $C_{RS}$ by using $\mathbf{U}(t-1), \mathbf{V}(t-1)$ and $\mathbf{W}_{RS}$ for $\mathcal{X}$.

- RegimeExtraction, which obtains a new regime $\mathbf{W}_{RE}$, focuses on current seasonality, by applying the gradient descent method to $\mathbf{U}(t-1), \mathbf{V}(t-1)$ and $\mathbf{W}_{RS}$ over the whole $\mathcal{X}$. The initialization with $\mathbf{W}_{RS}$ prevents $\mathbf{W}_{RE}$ from overfitting to $\mathcal{X}$ because temporal data, $\mathcal{X}$, can be sparse. It then computes $C_{RE}$ with $\mathbf{U}(t-1), \mathbf{V}(t-1)$ and $\mathbf{W}_{RE}$ for $\mathcal{X}$.

If $C_{RS}$ is less than $C_{RE}$, the best regime index $z$ becomes one of $\{1, \ldots, g\}$. Otherwise, $z = g + 1$, $\mathcal{W}(g \times s \times k)$ is extended to $\mathcal{W}((g+1) \times s \times k)$, and then $g$ is incremented.

## 4.1 Regime-Aware Gradient Descent Method

Given a new observation $\mathbf{X}(t)$, we consider obtaining the current factors, $\mathbf{U}(t), \mathbf{V}(t), \mathcal{W}(t)$ via the previous ones $\mathbf{U}(t-1), \mathbf{V}(t-1), \mathcal{W}(t-1)$ by minimizing error with respect to $\mathbf{X}(t)$. This allows the model to fit current dynamics smoothly while considering the past patterns. Thus, our algorithm performs the gradient descent method with a small learning rate $\alpha > 0$. The gradient step keeps the error with respect to $\mathbf{X}(t)$ low. Letting $z$ be the current regime index, and $\hat{\mathbf{X}}(t)$ be the predicted matrix of $\mathbf{X}(t)$ with the previous parameters:

$$\hat{\mathbf{X}}(t) = \mathbf{U}(t-1)\mathbf{W}^{(z)}(t-s)\mathbf{V}(t-1)^T, \tag{7}$$

the gradient update to $\mathbf{u}_i(t)$ and $\mathbf{v}_i(t)$ is given by:

$$\mathbf{u}_i(t) \leftarrow \mathbf{u}_i(t-1) + \alpha(\mathbf{X}(t) - \hat{\mathbf{X}}(t))\mathbf{v}_i(t-1)w_i^{(z)}(t-s), \tag{8}$$

$$\mathbf{v}_i(t) \leftarrow \mathbf{v}_i(t-1) + \alpha(\mathbf{X}(t) - \hat{\mathbf{X}}(t))^T\mathbf{u}_i(t-1)w_i^{(z)}(t-s). \tag{9}$$

The updated $\mathbf{u}_i(t)$ and $\mathbf{v}_i(t)$ are projected such that

$$\mathbf{u}_i(t) \leftarrow \max(0, \mathbf{u}_i(t)); \mathbf{v}_i(t) \leftarrow \max(0, \mathbf{v}_i(t)) \tag{10}$$

to keep the parameters non-negative. After all, the algorithm normalizes $\mathbf{u}_i(t)$ and $\mathbf{v}_i(t)$ by dividing by their norms $\|\mathbf{u}_i(t)\|$ and $\|\mathbf{v}_i(t)\|$, respectively:

$$\mathbf{u}_i(t) \leftarrow \mathbf{u}_i(t)/\|\mathbf{u}_i(t)\|; \quad \mathbf{v}_i(t) \leftarrow \mathbf{v}_i(t)/\|\mathbf{v}_i(t)\|, \tag{11}$$

and then $w_i^{(z)}$ multiplied by the two norms keeps the normalization constraint:

$$w_i^{(z)} \leftarrow w_i^{(z)}(t-s) \cdot \|\mathbf{u}_i(t)\| \cdot \|\mathbf{v}_i(t)\|, \tag{12}$$

which captures the seasonal adjustments for the $i$-th component.

---

**Algorithm 1** SSMF

---

**Input:** (a) New observation $\mathbf{X}(t)$ and previous factors $\mathbf{U}(t-1), \mathbf{V}(t-1), \mathcal{W}(t-1)$,
**Output:** Updated factors: $\mathbf{U}(t), \mathbf{V}(t), \mathcal{W}(t)$
1: /* Update the queue to keep the most recent season */
2: $\mathcal{X} \leftarrow \{\mathbf{X}(t-s+1), \ldots, \mathbf{X}(t)\}$;
3: /* (1) RegimeSelection */
4: $\mathbf{W}_{RS} \leftarrow \underset{\mathbf{W}' \in \{\mathbf{W}^{(1)}, \ldots, \mathbf{W}^{(g)}\}}{\arg\min} <\mathcal{X}; \mathbf{U}(t-1), \mathbf{V}(t-1), \mathbf{W}'>$;
5: /* (2) RegimeExtraction */
6: $\mathbf{W}_{RE} \leftarrow \underset{\mathbf{W}'}{\arg\min} <\mathcal{X}|\mathbf{U}(t-1), \mathbf{V}(t-1), \mathbf{W}'>$;
7: $C_{RS} = <\mathcal{X}; \mathbf{U}(t-1), \mathbf{V}(t-1), \mathbf{W}_{RS}>$; $C_{RE} = <\mathcal{X}; \mathbf{U}(t-1), \mathbf{V}(t-1), \mathbf{W}_{RE}>$;
8: **if** $C_{RS}$ **is less than** $C_{RE}$ **then**
9: $\quad \mathbf{W}^{(z)} = \text{diag}(\mathbf{W}_{RS}(t))$;
10: **else**
11: $\quad \mathbf{W}^{(z)} = \text{diag}(\mathbf{W}_{RE}(t))$; $\mathcal{W}(t) = \mathcal{W}(t-1) \cup \mathbf{W}_{RE}$; $g = g+1$;
12: **end if**
13: /* Perform gradient updates */
14: $\hat{\mathbf{X}}(t) \leftarrow \mathbf{U}(t-1)\mathbf{W}^{(z)}(t-s)\mathbf{V}(t-1)^T$;
15: $\mathbf{U}(t) \leftarrow \mathbf{U}(t-1) + \alpha(\mathbf{X}(t) - \hat{\mathbf{X}}(t))\mathbf{V}(t-1)\mathbf{W}^{(z)}(t-s)$;
16: $\mathbf{V}(t) \leftarrow \mathbf{V}(t-1) + \alpha(\mathbf{X}(t) - \hat{\mathbf{X}}(t))^T\mathbf{U}(t-1)\mathbf{W}^{(z)}(t-s)$;
17: **for** $i = 1 : k$ **do**
18: $\quad$ /* Project onto non-negative constraint */
19: $\quad \mathbf{u}_i(t) \leftarrow \max(0, \mathbf{u}_i(t))$; $\mathbf{v}_i(t) \leftarrow \max(0, \mathbf{v}_i(t))$
20: $\quad$ /* Renormalize all the factors */
21: $\quad w_i^{(z)} \leftarrow w_i^{(z)}(t-s) \cdot \|\mathbf{u}_i(t)\| \cdot \|\mathbf{v}_i(t)\|$;
22: $\quad \mathbf{u}_i(t) \leftarrow \mathbf{u}_i(t)/\|\mathbf{u}_i(t)\|$; $\mathbf{v}_i(t) \leftarrow \mathbf{v}_i(t)/\|\mathbf{v}_i(t)\|$;
23: **end for**
24: **return** $\mathbf{U}(t), \mathbf{V}(t), \mathcal{W}(t)$

---

### 4.2 Forecasting

To forecast $t$-steps ahead values for a given $t > r$, we use the latest factors, $\mathbf{U}(r)$, $\mathbf{V}(r)$, and $\mathbf{W}_{t_s}^{(z)}(r)$, where $t_s$ is the most recent season observed before time $r$, which corresponds to the season at time $t$. Namely, $t_s = r - (r - t \mod s)$. By choosing the best regime $z$, which gives the minimum description cost in the last one season, the predicted matrix for time $t$ is thus computed as follows.

$$\hat{\mathbf{X}}(t) = \mathbf{U}(r)\mathbf{W}_{t_s}^{(i)}(r)\mathbf{V}(r). \tag{13}$$

### 4.3 Theoretical Analysis

We show the time and memory complexity of SSMF. The key advantage is that both of the complexities are constant with regard to a given tensor length, even if it evolves without bound.

**Lemma 1** *The time complexity of SSMF is $O(sgkmn + sk^2(m+n))$ per time point.*

**Proof 1** *The algorithm first computes the model description cost between $\mathbf{X}(t)$ and $\hat{\mathbf{X}}(t)$ using $g$ regimes in $O(gkmn)$, which is repeated over $t = t-s+1, \ldots, r$. To obtain a new regime, it performs the gradient descent update to $\mathcal{X}$, which takes $O(s(kmn+k^2(m+n)))$. Then, for a selected regimes, it performs the gradient descent update, which requires $O(kmn + k^2(m+n))$. The projection to non-negative values and normalization require $O(m+n)$ for $k$ loops, thus $O(k(m+n))$ in total. Overall, the time complexity of SSMF is $O(sgkmn + sk^2(m+n))$ per time point.*

**Lemma 2** *The memory complexity of SSMF is $O(k(m+n+gs) + |\mathcal{X}|)$ per time point.*

**Proof 2** *SSMF retains the two consecutive factors, $\mathbf{U}$ and $\mathbf{V}$ with shape $(m \times k)$ and $(n \times k)$, respectively, which requires $O(k(m+n))$. For seasonal factors in $g$ regimes, it maintains a tensor $\mathcal{W}$ with shape $(g \times s \times k)$. It also needs a space to retain a sparse tensor, $|\mathcal{X}|$, which is the number of observed elements. Therefore, the total memory space required for SSMF is $O(k(m+n+gs) + |\mathcal{X}|)$ per time point.*

Table 2: Dataset Description

| Name | Rows | Columns | Duration | Sparsity | Frequency | $s$ | $r_{test}$ |
|---|---|---|---|---|---|---|---|
| NYC-YT | 265 | 265 | 4368 | 98.3% | Hourly | 168 | 500 |
| NYC-CB | 409 | 64 | 35785 | 96.2% | Hourly | 168 | 500 |
| DISEASE | 57 | 36 | 3370 | 92.4% | Weekly | 52 | 200 |

## 5 Experiments

In this section, we show how our proposed method works on real tensor streams. The experiments were designed to answer the following questions.

**Q1. Accuracy:** How accurately does it predict future events?

**Q2. Scalability:** How does it scale with input tensor length?

**Q3. Effectiveness:** How well does it extract meaningful latent dynamics/components?

We implemented our algorithm in Python (ver. 3.7.4) and all the experiments were conducted on an Intel Xeon W-2123 3.6GHz quad-core CPU with 128GB of memory and running Linux. Table 2 shows the datasets used to evaluate our method:

**NYC-YT.**[3] NYC Yellow Taxi trip records during the term from 2020-01-01 to 2020-06-30 in the pairs of 265 pick-up and drop-off locations.

**NYC-CB.**[4] NYC CitiBike ride trips from 2017-03-01 to 2021-03-01. We only considered stations that had at-least 50k rentals during the term, and extracted 409 stations. We then categorized users by age from 17 to 80 and obtained the stream of ride counts at the 409 stations.

**DISEASE.**[5] A stream of the weekly number of infections by 36 kinds of diseases at the 57 states/regions in the US, which were collected from 1950-01-01 to 2014-07-27.

### 5.1 Accuracy

We first describe a comparison in terms of forecasting using Root Mean Square Error (RMSE). Since [Hooi et al., 2019] outperforms several baselines including CPHW and TSVDCWT, we omit the baselines from our experiments. Our baseline methods include (1) matrix/tensor factorization approaches: NCP [Xu and Yin, 2013]: a non-negative CP decomposition, optimized by a block coordinate descent method for sparse data. (2) SMF [Hooi et al., 2019]: an online algorithm for seasonal matrix factorization, which manages a single seasonal factor. (3) TRMF [Yu et al., 2016]: a method to take into account temporal and seasonal patterns in matrix factorization.

For each method, the initialization process was conducted on the first three seasons of an input tensor. For TRMF, we searched for the best three regularization coefficients, $\lambda_I$, $\lambda_{AR}$, $\lambda_{Lag}$, in $\lambda = \{0.0001, 0.001, 0.01, 0.1, 1, 10\}$. Auto-coefficients of TRMF are set for at time points $r - 1, \ldots, r - \lceil s/2 \rceil$ as well as at three time points $r - s, r - (s + 1), r - (s + 2)$ to consider seasonality. For SSMF and SMF, we determined a learning rate $\alpha$ in $\alpha = \{0.1, 0.2, 0.3, 0.4\}$ by cross-validation in each training data. The number of components $k$ was set to 15 among all these methods for a fair comparison. After the initialization, each algorithm takes the first $r_{train}$ time steps, and forecasts the next $r_{test}$ steps, as defined in Table 2. This procedure was repeated for every $r_{test}$ steps, for example, $r_{train} = \{500, 1000, 1500, \ldots\}$ in NYC-YT dataset.

Figure 2 shows the results of RMSE on all three datasets, where SSMF consistently outperforms its baselines. We observe that our method shows a better improvement on (b) NYC-CB than (a) NYC-YT, whereas the two datasets have similar characteristics because detecting regime shifts is more effective in forecasting in a longer data duration. DISEASE dataset contains repetitive regime shifts caused by pandemics and developments of vaccines, therefore, SSMF performs forecasting effectively.

---

[3]`https://www1.nyc.gov/site/tlc/about/tlc-trip-record-data.page`
[4]`https://s3.amazonaws.com/tripdata/index.html`
[5]`https://www.tycho.pitt.edu/data/`

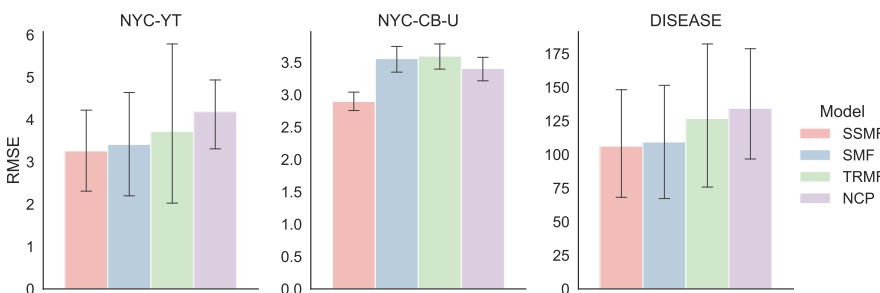

Figure 2: Forecasting performance in terms of RMSE: SSMF consistently outperforms the baselines thereby showing that regime shift detection adapts the model to recent patterns more quickly and effectively than the baselines.

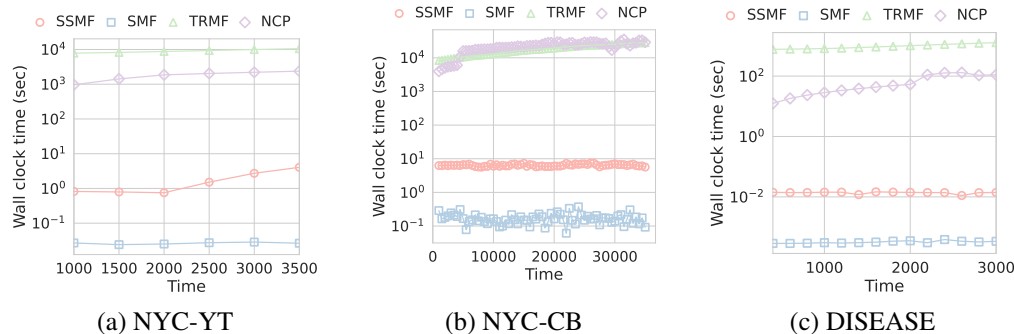

| (a) NYC-YT | (b) NYC-CB | (c) DISEASE |

Figure 3: Wall clock time vs. data stream length: SSMF efficiently monitors data streams to find regimes.

## 5.2 Scalability

We discuss the performance of SSMF in terms of computational time on the three datasets. Figure 3 shows the wall clock time vs. data stream length, where the experimental setting is same as of subsection 5.1. The figures show that our proposed method is scalable for mining and forecasting large sparse streams. Although the scalability of SSMF is linear along the number of regimes, the growth rate of its running time is greatly less than that of the other offline baselines. This is because the proposed method can summarize distinct seasonal patterns as regimes effectively. SMF is the fastest algorithm but it does not consider regime shifts in online processing. As described in the accuracy comparison section, our method is more effective when data streams have significant pattern shifts, which cannot be captured only with smooth updates on a single, fixed component.

## 5.3 Real-world Effectiveness

In this section, we show an example of the real-world effectiveness of our approach using the NYC-YT dataset. Figure 4 shows the overview of the two representative regimes detected by SSMF, where we selected two of the fifteen components to describe. The bottom bar represents the timeline of regime detection. SSMF kept using the initial regime during the blue term, while it generated new regimes during the red term. The three rows of plots are the visualization of $\mathbf{W}$, $\mathbf{U}$, $\mathbf{V}$, where the first two columns correspond to the blue regimes, and the next two to the red.

In mid March 2020, New York City's governor issued stay-at-home orders, and SSMF detected the event as the regime shifts to adapt to the changed demands of taxi rides. Separating regimes and learning individual patterns help us to interpret the results as follows: Component 1 of the blue regime shows a seasonal pattern that follows a weekday/weekend pattern. According to the above two maps, more taxi rides are concentrated around Manhattan before the pandemic. In the red regime, however, the strength of the seasonal component became about 10% weaker than the one before. Accordingly, the frequent pickup and dropoff became smaller. This area has a large number of bars and restaurants, suggesting that this component corresponds roughly to entertainment-related trips, whose demands

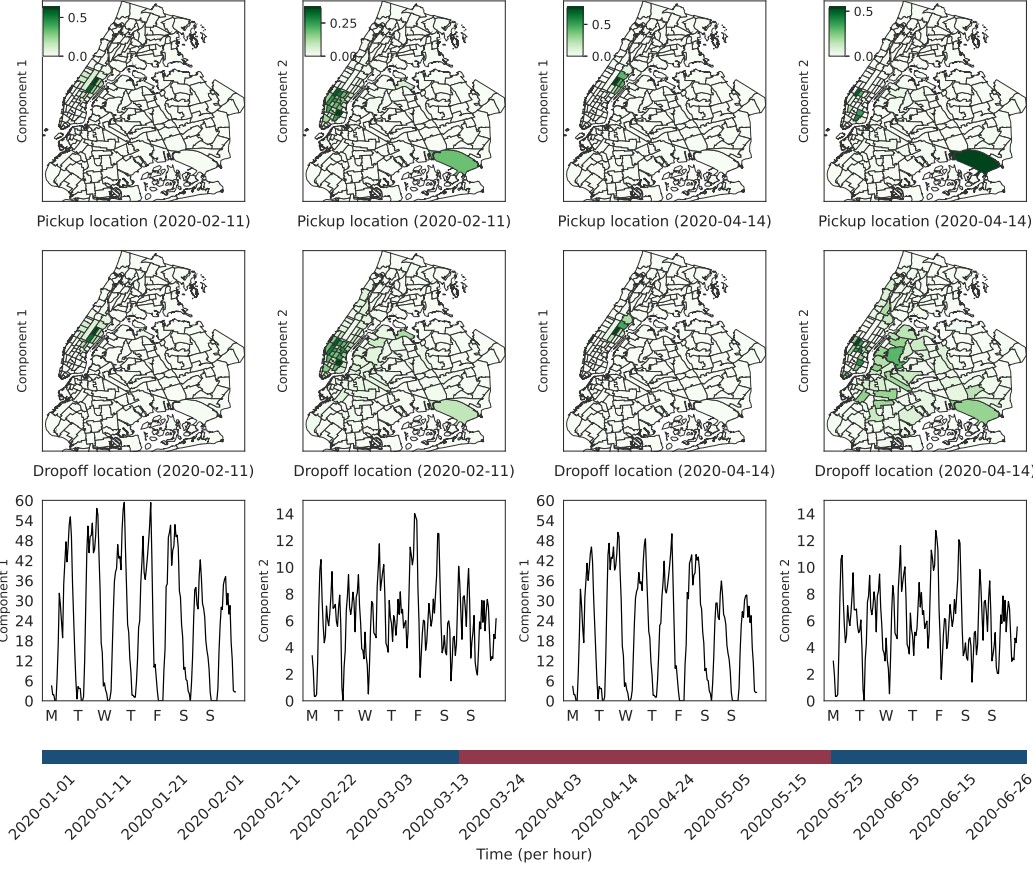

Figure 4: SSMF can automatically detect regime shifts caused by a pandemic based on multiple seasonal factors: Component 1, corresponding roughly to entertainment, decays in the pandemic, while component 2, representing trips between Manhattan and Kennedy Airport, experiences a shift toward trips from the airport to wide areas of New York City.

are strongly affected by the pandemic. Component 2, in contrast, is concentrated between Manhattan and Kennedy Airport, suggesting that this component captures trips to and fro the airport. Under the pandemic situation, this component shifted toward trips from the airport to destinations scattered around New York City. This observation could suggest that customers arrived at the airport, and then used a taxi to avoid public transit. After this phase, our method switched to the first regime again, returning to the previous seasonal pattern. The demand for taxi rides began to recover, which roughly aligns with reopening in New York City, which began in early June. Consequently, regime shifts allow us to distinguish significant changes in seasonal behavior as well as contextual information between attributes. In online learning, the mechanism also supports for quick adaptation and updates, which contributes to improving forecasting accuracy.

## 6    Conclusion

In this paper, we proposed Shifting Seasonal Matrix Factorization (SSMF) method, for effectively and efficiently forecasting future activities in data streams. The model employs multiple distinct seasonal factors to summarize temporal characteristics as regimes, in which the number of regimes is adaptively increased based on a lossless data encoding scheme. We empirically demonstrate that on three real-world data streams, our forecasting method SSMF has the following advantages over the baseline methods: (a) By using the proposed switching mechanism in an online manner, SSMF *accurately* forecasts future activities. (b) *Scales* linearly by learning matrix factorization in constant time and memory. (c) Helps understand data streams by *effectively* extracting complex time-varying patterns.

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
