# OpenReview forum: "SSMF: Shifting Seasonal Matrix Factorization"
_NeurIPS.cc/2021/Conference — NeurIPS 2021 Poster_

### Official Review · Reviewer_ZNwm · 2021-07-16

**Rating:** 7
**Confidence:** 3

**Summary:**

The authors develop a new online tensor factorization method for real-time forecasting. The propose approach is designed for dealing with seasonality patterns and automatically switch between different regimes. The authors have shown this method achieves better accuracy compared to existing methods in forecasting taxi/bike rides and infections. The method has shown to be scalable in terms of its memory and test costs.

**Limitations And Societal Impact:**

The authors have proposed a general matrix factorization method that can automatically identify and switch between different regimes. This is an important property for many time series data sets.

**Main Review:**

Overall, I think this is a well-organized paper and the content is solid. I just have a few comments
1. The seasonality patterns are important for many real-world applications, especially in climate and environment modeling. It would be better if the authors could discuss how easy this proposed algorithm can be generally applied to other applications.
2. The proposed method is shown to outperform several MF-based baselines. I am wondering if the authors can also compare it with deep learning-based algorithms.
3. In many real-world scenarios, we may have access to when the regime shift occurs (e.g., based on seasonality, human interventions, or other separate analysis). Would the proposed method be able to leverage such information? Also, if we have already known such time partitions, would the proposed method outperform other algorithms, e.g., domain-adaptation?


**Time Spent Reviewing:**

2

---

> ### Author Response · Authors · 2021-08-18
> **Author Response**
>
> We greatly appreciate the positive and insightful comments.
>
> - More discussion of applications: Thanks for the suggestion. Indeed, seasonal patterns are ubiquitous in many time series applications, including climate, environment, industrial, biological etc. sensor data, and our algorithm can readily be applied to any such application with matrix-structured data as input: for example, sensor data organized by $\textrm{sensor type} \times \textrm{location}$, like in our Disease dataset example. We plan to add more such discussion in the final version.
>
> - We agree with the suggestion of incorporating deep learning-based algorithms as additional baselines - though our current draft focuses on matrix factorization-type approaches, we plan to add this to future versions of the paper.
>
> - Yes, it would be able to leverage such information. The case where we know when the regime shifts occur in advance is effectively a special case of our framework, which can be handled by straightforwardly omitting the algorithm's selection of the regime shifting time whenever it is known through prior knowledge. As for whether the resulting approach would outperform baselines such as domain adaptation, we believe that this would depend on whether the data contains patterns such as seasonality or dynamics that can be well-represented by matrix components, thus allowing our method to perform well. A set of several regimes pre-trained by such domain knowledge can be also used for the algorithm, in which it monitors how subsequent data streams evolve based on the regimes.

---

### Official Review · Reviewer_gthx · 2021-07-18

**Rating:** 5
**Confidence:** 4

**Summary:**

This paper aims to solve the problem of forecasting future events given a data stream of events. In order to achieve this, the authors propose a method called Shifting Seasonal Matrix Factorization, which can detect the regime shifts in seasonal patterns, and work in online settings. The authors claimed that, the advantages of the proposed method over previous methods are its capability to dealing with sparse data, regime shifts, seasonal patterns, and drifting components.

The authors conduct experiments on three datasets, NYC-YT, NYC-CB and DISEASE. All the methods are compared in their capability of predicting the events. The authors also show a case study in NYC to show the results of detected events.

**Limitations And Societal Impact:**

No separate Limitations and Societal Impact part is provided.

The authors may consider how these event predictions can help with dealing with traffic congestion, reducing the time that people need to get a taxi, or pandemic intervention.

**Main Review:**

Strength:
1. The authors are working on an interesting problem. Considering data sparsity, seasonality and drifting is an interesting and important problem in the matrix factorization problem.
2. The description of the method is clear and easy to follow. Readers can easily understand the algorithm through the descriptions in Section 3 and 4.
3. The source code is provided, which can be used for replicating the results.

Weakness:
1. The compared methods are not comprehensive enough and out-of-dated. The authors have mentioned several more advanced baselines in the related work section, e.g., [Takahashi et al., 2017], [Song et al., 2017], [Najafi et al., 2019]. It is not clear why these methods are not compared. In addition, the investigated problem is also highly related to time series prediction. It is not clear why these methods are not compared.

2. The illustration of the case study is not straightforward. For the maps, the authors may consider zooming in and showing more cases. The current examples can only show the changes around JFK area. The authors may also adjust the axis of the time series to better illustrate the change in volume.

3. The authors claim the capability of dealing with data sparsity. Hence, it is preferred if some experiments can be conducted on dealing with different sparse levels.

**Time Spent Reviewing:**

3

---

> ### Author Response · Authors · 2021-08-18
> **Author Response**
>
> We greatly appreciate the detailed and insightful comments.
> - Why several more advanced baselines are not compared?: [Song et al., 2017] and [Najafi et al., 2019] are streaming tensor completion methods, but are not be able to consider temporal patterns including seasonality. Instead, we chose to compare with the most directly relevant and similar methods, including SMF and TRMF, which are both time series matrix factorization approaches which allow for seasonal patterns. We additionally plan to improve the description of these baselines and their comparison to our method, including in Table 1.
>
> - The illustration of the case study is not straightforward: We agree and greatly appreciate the suggestions for the changes to the figure, and will use them to improve the clarity of the figure and the case study.
>
> - Some more experiments conducted on different sparse levels: This is indeed a helpful addition - we plan to add them to future versions of the paper.

---

> > ### Comment · Reviewer_gthx · 2021-08-18
> > **What is the performance of the baselines without considering seasonality?**
> >
> > The authors have mentioned that several baselines fail to consider the seasonality, and that is why they are not compared. Actually, it is still helpful to compare with them although they have not addressed all the information.
> >
> > Thanks for the response to the other comments.

---

### Official Review · Reviewer_Fqt5 · 2021-07-19

**Rating:** 7
**Confidence:** 3

**Summary:**

This paper studies online tensor factorization for real-time event forecasting with seasonality. The key to solve the event forecasting problem here is to detect the seasonal pattern shift and adapt to the shift for an accurate  event forecasting. The closest related work is SMF [Hooi et al., 2019], which is an online forecasting model based on matrix factorization and taking as well seasonal patterns into account. SMF differs from the proposed SSMF by considering only a single seasonal factor, whereas SSMF model seasonal shifts (regimes) individually.



**Main Review:**

The designed regime selection and extraction mechanism enables the prediction model to have stronger adaptability to the seasonal shifts. The experimental results show that the proposed SSMF model outperforms all baselines. An interesting demonstration example is also provided to demonstrate the capability of SSMF on automatically detecting regime shifts caused by a pandemic.   In general, this work is a valuable contribution to the field of event forecasting.

Authors are suggested to carefully check the writing errors: “how well a given model compress ” and “a x” and so on.


**Time Spent Reviewing:**

10

---

> ### Author Response · Authors · 2021-08-18
> **Author Response**
>
> We greatly appreciate the positive and insightful comments. We have corrected the writing errors in the final version.

---

### Official Review · Reviewer_CeEZ · 2021-07-20

**Rating:** 6
**Confidence:** 4

**Summary:**

The work aims to solve the problem of forecasting for incomplete highly-dynamic time-series data with evolving seasonal patterns (regimes). It uses clever techniques to detect whether seasonal shift occurred in new data and updates the model with the corresponding new regime if necessary. Model training is performed in an online fashion over a data stream and is based on an incremental matrix factorization with adaptive selection of the most appropriate regime. It utilizes minimum description length principle for the task. The authors provide a straightforward learning scheme and demonstrate advantages of the proposed approach on several realistic datasets.

**Limitations And Societal Impact:**

I believe that the work doesn't possess any components that may directly lead to negative societal impacts.

**Main Review:**

*Originality*:
The authors make a good work on positioning their approach among other works and describing the essential differences. The idea of adaptive regimes update/selection is both interesting and novel. The way the task of regime selection is solved with the MDL principle seems to be very appropriate.

*Quality*:
The proposed approach is clearly described and the all necessary part for its (re)implementation seem to be in place. Both time and space complexities are provided. The scalability analysis performed by the authors is convincing. However, there are a few issues.

Section 3.2, equation between line 135 and 135: the formulation is based on the number of non-zero elements, which, I believe, implicitly assumes sparse matrices/tensors (with non-negative elements). However, eq. (4) seems to be free of such assumptions. Isn't there a contradiction? If there are additional assumptions on relaxation of any constraints, it would be good to have them explicitly described. Otherwise, the choice of a distribution should be different.

Section 5.1, line 232: I'm not convinced that fixing the same number of components ensures fair comparison. Different methods are likely to require different values of k for achieving the best quality. If there are certain memory constraints, the number of components must be selected w.r.t. the space complexity of each method. If there are any technical limitations for this, then at least he trendline for different k should be demonstrated.

*Clarity*:
Section 2.1 is named *Matrix Factorization*, but talks a lot about tensor factorization as well, which add to the impression of text inconsistency. The authors mix together these terms.
Moreover, the term SVD is used quite broadly in the text (e.g., line 62). However, it is related to a very specific matrix decomposition method. The authors are probably influenced by the recommender systems literature, with many matrix factorization techniques carrying SVD in their names (e.g., FunkSVD, SVD++, AsySVD, etc.). I'd argue against such terminology as, in general, matrix factorization methods do not provide the same set of algebraic properties as in the case of SVD.

In Section 2.2, line 80, what does the term *non-linear dynamical system* refer to? It is not clear, why this term is used to contrast other approaches against the proposed one. I couldn't find any clarification on that further in the text.

In Section 3, Table 2 is not very helpful and creates a lot of whitespace. It is also excessive. The first two rows are describing the same object X(t). It would be probably better to just introduce proper notation directly in the text.

*Significance*:
The proposed approach open many perspective for further improvements and modifications. Given its relative simplicity, it may become a source for fruitful future research.

*Minor issues*:
- line 98: "and r evolves" -> "as r evolves" ? The notation with r = r + 1 is also a bit cumbersome.
- line 51: The link to Dropbox should be explicit, e.g., via footnote.
- line 35: unless we talk about very special cases, they almost never converge to optimal parameters and are likely to find only a local optimum. I'd propose to rephrase the sentence. Probably, it's just enough to leave the second part saying about the lower accuracy.
- line 137: compress -> compresses
- line 134: provide -> provided

**Time Spent Reviewing:**

7

---

> ### Author Response · Authors · 2021-08-18
> **Author Response**
>
> We greatly appreciate the positive and insightful comments.
> - (Quality) Assumption of the distribution for error term:
> Thanks for the interesting question! The reason there is no contradiction between the formulation of factors/components and the distribution of errors is that the former relates to the encoding cost for $\mathbf{U}$ and $\mathbf{V}$, while the latter relates to the encoding cost of $\mathbf{X}$ given the factors, which can be modelled largely separately from one another (aside from their encoding costs being added at the end). At the same time, we agree that other distributions, e.g., Poisson, can lead to a more general and possibly effective approach. Hence, although our current work focuses on the simplest case, we plan to consider such extensions for future work.
>
> - (Quality) Comparison with fixed the number of components: Thank you for the comment. Our choice of using $k=15$ was motivated by the same choice being made in the closest related work SMF (Hooi et al., 2019) and its own baselines. Thus, following this choice allows us to provide a more direct and straightforward comparison with their paper, and ensures that we are running their method based on the author's suggested parameters, and to fairly assess the performance of regime switching, i.e., which learning approach is more effective between online update (not forgetting previous patterns) and online update with switching (forgetting previous patterns to learn new patterns). At the same time, we do plan to provide additional results across a range of different values of $k$ in future versions of the paper.
>
> - (Clarity) What does the term non-linear dynamical system refer to?: The non-linear dynamical system is a family of state space models, such as the one used in RegimeCast (Matsubara and Sakurai, 2016), which is one of the algorithms to handle regime switching. We refer to it to compare and contrast our model with RegimeCast, as in Table 1, noting that while it can model regime shifts, it cannot capture dynamics in sparse time series.
>
> - Minor Issues: Thank you for your suggestions and corrections! We have corrected the typos in the final version.

---

### Decision · Program_Chairs · 2021-09-28

**Decision:**

Accept (Poster)

**Comment:**

The paper studies forecasting of seasonally varying time series using matrix factorisation of a shifting tensor stream. The proposed algorithm, SSMF, makes forecasts by identifying the current regime and applying the corresponding sub-model. SSMF is analysed with respect to time and memory complexity per time point. The approach is evaluated with respect to scalability and real-world effectiveness in a series of empirical experiments.

The reviwers appreciated the novelty of the formulation and the importance of the problem. They deemed the paper well written and the results sound. Some concerns were raised regarding the clarity of the paper which were addressed in the author response. Finally, two reviewers asked for comparisons with additional baselines, but I understand the choice to focus on matrix factorisation methods in this setting. Overall, most of the reviewers were in favour of accepting the paper and no major issues remained after the discussion phase.

**Consistency Experiment:**

NeurIPS has a long history of experimentation. In 2014, NeurIPS ran an experiment in which 10% of submissions were reviewed by two independent committees to quantify the randomness in the review process. This year, we repeated a variant of this experiment to see how the quality of the review process has changed over time.  This paper was part of the experiment and was therefore assigned to two committees (consisting of reviewers, an Area Chair, and a Senior Area Chair) that reached independent decisions.  If both committees made the same recommendation, this recommendation was followed. If a single committee recommended acceptance, the paper was accepted (with the exception of a few cases in which the other committee identified what we considered a fatal flaw, e.g., an error in a key result).

This copy’s committee reached the following decision: **Accept (Poster)**

The other committee assigned to the paper recommended **Reject**.  You can find the other set of reviews, along with any follow up discussion with the authors here:
https://openreview.net/forum?id=AqprMSXI1Wn